

# Zero-inflated models for the evaluation of colorectal polyps in colon cancer screening studies—a value-based biostatistics practice

Alok K. Dwivedi[1], Sherif E. Elhanafi[2], Mohamed O. Othman[3] and Marc J. Zuckerman[2]

[1] Division of Biostatistics & Epidemiology, Department of Molecular and Translational Medicine, Paul L. Foster School of Medicine, Texas Tech University Health Science Center, El Paso, Texas, United States
[2] Division of Gastroenterology, Department of Internal Medicine, Paul L. Foster School of Medicine, Texas Tech University Health Science Center, El Paso, Texas, United States
[3] Gastroenterology and Hepatology Section, Baylor College of Medicine, Houston, Texas, United States

Corresponding author
Alok K. Dwivedi,
alok.dwivedi@ttuhsc.edu

## ABSTRACT

**Background:** Colon cancer screening studies are needed for the early detection of colorectal polyps to reduce the risk of colorectal cancer. Unfortunately, the data generated on colon polyps are typically analyzed in their dichotomized form and sometimes with standard count models, which leads to potentially inaccurate findings in research studies. A more appropriate approach for evaluating colon polyps is zero-inflated models, considering undetected existing polyps at colonoscopy screening.

**Method:** We demonstrated the application of the zero-inflated and hurdle models including zero-inflated Poisson (ZIP), zero-inflated robust Poisson (ZIRP), zero-inflated negative binomial (ZINB), zero-inflated generalized Poisson (ZIGP), zero hurdle Poisson (ZHP), and zero hurdle negative binomial (ZHNB) models, and compared them with standard approaches including logistic regression (LR), Poisson regression (PR), robust Poisson (RP), and negative binomial (NB) regression for the evaluation of colorectal polyps using datasets from two randomized studies and one observational study. We also facilitated a step-by-step approach for selecting appropriate models for analyzing polyp data.

**Results:** All datasets yielded a significant amount of no polyps and therefore inflated or hurdle models performed best over single distribution models. We showed that cap-assisted colonoscopy yielded significantly more colon polyps (risk ratio [RR] = 1.38; 95% confidence interval [CI] [1.05–1.81]) compared with the standard colonoscopy by using the ZIP analysis. However, these findings were missed by standard analytic methods, including LR (odds ratio [OR] = 0.90; 95% CI [0.59–1.37]), PR (RR = 1.14; 95% CI [0.93–1.41]), and NB (RR = 1.16; 95% CI [0.89–1.51]) for evaluating colon polyps. The standard approaches, such as LR, PR, RP, or NB regressions for analyzing polyp data, produced potentially inaccurate findings compared to zero-inflated models in all example datasets. Furthermore, simulation studies also confirmed the superiority of ZIRP over alternative models in a range of datasets differing from the case studies. ZIRP was found to be the optimal

method for analyzing polyp data in randomized studies, while the ZINB/ZHNB model showed a better fit in an observational study.

**Conclusion:** We suggest colonoscopy studies should jointly use the polyp detection rate and polyp counts as the quality measure. Based on theoretical, empirical, and simulation considerations, we encourage analysts to utilize zero-inflated models for evaluating colorectal polyps in colonoscopy screening studies for proper clinical interpretation of data and accurate reporting of findings. A similar approach can also be used for analyzing other types of polyp counts in colonoscopy studies.

## INTRODUCTION

Colorectal cancer (CRC) is the second leading cause of cancer-related mortality and the third most common cancer in the world. The estimated number of new cases of CRC is highest in the United States (*Xi & Xu, 2021*). CRC more often begins with a polyp, a noncancerous progression that develops in the large intestine. Some types of polyps have the potential to develop CRC if left untreated (*Helsingen & Kalager, 2022*). The leading precursor lesion of CRC is adenomatous polyps (*Klos & Dharmarajan, 2016*). Although CRC is a major health concern, declining trends have been observed in the incidence and mortality rates in the United States due to the increased use of colonoscopies for CRC screening and early treatment of polyps (*Kusnik et al., 2023*). Continuous advances in colonoscopy procedures are being made to improve the diagnosis, early detection, and treatment of polyps among at-risk populations (*Pamudurthy, Lodhia & Konda, 2020*). The performance matrix of colonoscopy studies typically includes the detection of polyps or adenoma counts (*Amano et al., 2018*; *Delavari et al., 2015*; *Winawer et al., 2006*). Unfortunately, these outcomes in colonoscopy studies are analyzed insufficiently and potentially inaccurately.

Since researchers are often interested in evaluating polyp detection rate (PDR) in colonoscopy studies, the number of polyps is analyzed in their dichotomized form, which may produce insufficient findings (*Liu et al., 2020*; *Rutter et al., 2020*). In statistical parlance, analyzing data deviated from the original form of data collection typically causes information loss and produces inefficient results (*Altman & Royston, 2006*; *Dwivedi, 2022*). Many research studies are available for evaluating colorectal polyps in colonoscopic studies. However, most of these studies analyzed colorectal polyps in their dichotomized form (*Nazarian et al., 2021*; *Pan et al., 2020*). A study highlighted the use of count data models for the analysis of the number of adenomatous polyps in a phase III randomized trial. This study demonstrated how critical it is to analyze the number of adenoma polyps instead of their categorized form (*Xie & Aickin, 1997*). A study established that the count of polyps follows a Poisson distribution (*Emerson et al., 1993*). Recently, a few studies applied the Poisson regression (PR) model to analyze the number of polyps in colonoscopic studies (*Drew et al., 2016*; *Liu et al., 2020*). However, a large proportion of

colonoscopic data yields no polyps, leading to an overdispersion issue, *i.e.*, observed variance is greater than the estimated variance from the model in analyzing polyp data (*Davies et al., 2023*; *Hull et al., 2019*; *Singh et al., 2017*). The overdispersion issue in the standard PR model can be addressed by using a robust variance estimation approach in the PR model using a quasi-Poisson (QP), robust Poisson (RP), overdispersion Poisson, negative binomial (NB), or generalized Poisson (GP) model instead of a standard Poisson model (*Harris, Yang & Hardin, 2012*; *Payne et al., 2018*). However, due to an excess amount of zeros (no polyps), these single-distribution models may not be reliable enough to produce accurate findings (*Campbell, 2021*). Although clinicians are often interested in PDR estimation and comparison in the primary analysis of colonoscopic studies, which cannot be directly obtained by simple count data models, these can be estimated indirectly *via* computing marginal effect sizes, which are typically useful in clinical trial studies.

While evaluating the count data between different groups, analysts sometimes use the logistic regression (LR) model by categorizing data into a binary form. However, this approach does not demonstrate whether groups are significantly different in detecting the extent of counts. Generally, analysts apply standard count models such as PR, RP, GP, or NB models to evaluate count differences between groups. However, this approach produces a potentially inaccurate effect size as well as inadequate conclusions in the presence of excess zeros (*Fernandez & Vatcheva, 2022*; *Slymen et al., 2006*). Therefore, most trials evaluating polyp count data using these methods might have yielded potentially inaccurate conclusions or inaccurate effect sizes, affecting scientific progress and planning of future trials. As per value-based biostatistics practice, multiple questions should be addressed simultaneously if the data structure allows (*Dwivedi, 2022*). Accordingly, we propose to analyze polyp data using mixed distribution models such as zero-inflated or zero hurdle count models, which would not only account for overdispersion due to excess zeros but also facilitate clinical interpretation of PDR along with the estimation and comparison of the extent of detected polyps. To the best of our knowledge, as of December 12, 2024, a quick search using the terms "zero-inflated" or "zero hurdle" on Google Scholar yielded 17,900 results, while these searches on PubMed yielded 2,565 results. Although inflated and hurdle methods are progressively used in medical research (*Haslett et al., 2022*), these methods have not been used for analyzing studies with polyp count data. There are multiple types of inflated or hurdle models, including zero-inflated Poisson (ZIP), zero-inflated robust Poisson (ZIRP), zero-inflated negative binomial (ZINB), zero-inflated generalized Poisson (ZIGP), zero hurdle Poisson (ZHP), and zero hurdle negative binomial model (ZHNB). The choice of hurdle and inflated models depends on the type of excess zeros, while the choice of specific distribution in the inflated or hurdle models depends on the over or under, or equidispersion outcome data and goodness of the fit of a specific model (*Dwivedi et al., 2010*, *2014b*). After eliminating excess zeros from the response, if the variance of the count data is equal to its mean, then it is called equidispersion, otherwise overdispersion (mean < variance) or underdispersion (mean > variance). We demonstrate the significance of using inflated models in analyzing polyp data using our three published studies (*Avalos et al., 2020*; *Elhanafi et al., 2017*; *Othman et al., 2017*) as well as the step-by-step approach for selecting an appropriate model.

## MATERIALS AND METHODS

In this report, we used published studies to demonstrate the utility of inflated models and the interpretation of data in analyzing polyp response yielded from either observational or experimental studies. Two of the three example datasets in this current study were generated from a randomized clinical trial.

### Example dataset 1

The first dataset was utilized from a randomized clinical trial that included 425 patients to compare the performance of cap-assisted colonoscopy (CAC) and standard colonoscopy (SC) for evaluating PDR (*Othman et al., 2017*). This trial aimed to demonstrate the efficacy of CAC in detecting increased PDR, adenoma detection rate (ADR), advanced adenoma detection rate (AADR), and large polyp size compared to the SC. This trial included covariates such as age, body mass index (BMI), sex, race, preparation quality, prior abdominal surgery, family history of colon cancer, colonoscopy indication, endoscope use, withdrawal time, procedure time, and cecal intubation time. Owing to randomization, these baseline covariates were balanced between randomized groups. However, age, sex, and total procedure time are typical predictors for PDR (*Shaukat et al., 2009*; *Taber & Romagnuolo, 2010*), and therefore, we adjusted for age, sex, and total procedure time while evaluating the differences between CAC and SC. We also validated results without any covariate adjustment in the analysis.

### Example dataset 2

The second dataset was utilized from another randomized clinical trial on 311 patients. This study compared the efficacy of a standard withdrawal time protocol (at least 6 min of withdrawal regardless of the side of the colon) *vs.* a segmental withdrawal time protocol (at least 3 min spent on the right side of the colon) for evaluating PDR, ADR, AADR, and large polyp size (*Avalos et al., 2020*). The baseline covariates collected in this study were age, BMI, sex, ethnicity, fellow present, colonoscope used for adult or pediatric, preparation quality, cecal intubation, withdrawal time, procedure time, and cecal intubation time. We analyzed data after adjusting for age, sex, and total procedure time. In addition, we also validated findings without any covariate adjustment.

### Example dataset 3

The third example dataset was extracted from an observational study. The aim of this study was to assess the impact of the initiation of a new gastroenterology (GI) fellowship program on PDR, ADR, and ADDR compared to the period without a GI fellowship program by using data on 2,127 patients who underwent screening colonoscopies (*Elhanafi et al., 2017*). However, we analyzed 1,936 patients owing to missing data on BMI. The covariates were age, BMI, sex, ethnicity, fellow present, time of the day, sedation type, preparation quality, mass present, and diverticulosis. Owing to the observational study design, these covariates were also included in our data models. We only retained variables in the final models that were associated with PDR or polyp counts. We demonstrated the differences in factors associated with PDR or polyp counts using standard approaches

compared with the final inflated model. We also validated findings after imputing missing data on BMI.

## Statistical models

### Logistic regression

The standard approach for analyzing polyp data is in a binary form, that is no polyps *vs*. detected polyps. The common model for analyzing binary outcome data is the LR model. LR is a linear relationship of covariates with log odds of the outcome. The exponentiated coefficient obtained from the LR yields an odds ratio (OR). To understand the factors associated with the detected polyps/adenomas, the LR model can be expressed as:

$$\log \text{(odds of detected polyps)} = \beta_0 + \beta_1 X_1 + \beta_2 X_2 + \ldots + \beta_n X_n$$

where, $\beta_0$ is the intercept, and $\beta_1$, $\beta_2$, …, $\beta_n$ are the coefficients associated with covariates ($X_i$ where i = 1 to n; n is the number of covariates).

### Count regression

Due to the loss of information by categorizing the number of polyps into a binary form, sometimes analysts use a count model to analyze the original form of data collection on colon polyps. The most common count model approach for analyzing polyp data is the PR model. However, QP, GP, or NB models are preferred in the presence of non-equidispersed data. The PR, QP, GP, or NB models fit the linear relationship between the log of the average number of polyps and covariates (*Drew et al., 2016*). The exponentiated coefficient obtained from the PR/NB/QP/GP regression yields risk ratio (RR), although sometimes referred to as incidence rate ratio (IRR), particularly in cohort or clinical trial studies (*Dwivedi et al., 2010*, *2014a*). The PR, QP, GP, or NB model can be expressed for modeling the expected (E) counts of detected polyps (*Ismail & Jemain, 2007*) as:

$$\log(E \text{ (number of detected polyps)}) = \beta_0 + \beta_1 X_1 + \beta_2 X_2 + \ldots + \beta_n X_n$$

where $\beta_0$ is the intercept and exponentiating this coefficient represents the expected count of polyps when all covariates are equal to zero, and $\beta_1$, $\beta_2$, …, $\beta_n$ are the coefficients associated with covariates ($X_i$ where i = 1 to n; n is the number of covariates). If the model aims to draw inference, then we may prefer using QP or RP, which is basically a PR, but the variances are estimated using quasi-likelihood methods or the Huber-Sandwich estimator method (*Slymen et al., 2006*; *Ver Hoef & Boveng, 2007*).

### Zero-inflated models

Inflated models are typically recommended for analyzing datasets that involve a large proportion of one count, usually zero, along with observed values. In this approach, the zeros are considered inflated or excess, which are not explained by the standard count models. It uses a two-part model or mixed distribution model, in which the distribution of counts is analyzed with a Poisson, NB, or GP regression model while the excess zeros *vs*. remaining counts are analyzed with an LR model (*Zafakali & Ahmad, 2013*). Since polyps

or adenomas are a precursor of CRC, no polyps are typically much larger in proportion than the remaining counts of polyps, recommending the use of mixed distribution models (*Helsingen & Kalager, 2022*). No polyps may also include undetected polyps that are missed at colonoscopy (*Emerson et al., 1993*; *Pamudurthy, Lodhia & Konda, 2020*), producing a mixture of zeros. Therefore, a zero-inflated model is more appropriate for describing the number of polyps regardless of its fit compared to the standard count models (*Fernandez & Vatcheva, 2022*). Moreover, the two components of the model allow clinicians to compare factors associated with both the likelihood of detected polyps and the extent of the number of polyps. The zero-inflated models jointly model the excess zeros as well as the number of observed values, and it can be expressed into two parts, including the excess zero and count components (*Dwivedi et al., 2010*):

$$P(Y = y_i) = \begin{cases} p_i + (1 - p_i) * f(y_i) & \text{if } y_i = 0 \\ (1 - p_i) * f(y_i) & \text{if } y_i > 0 \end{cases}$$

where $f(y_i)$ can be described with Poisson, NB, or GP distributions, and $p_i$ can be described with logit or probit models for the $i^{th}$ subject.

Excess zero component: The LR for excess zeros models the probability of observing excess zero counts (*i.e.*, the absence of polyps):

$$\log(\text{odds of excess no polyps}) = b_0 + b_1 X_1 + b_2 X_2 + \ldots + b_n X_n$$

where, $b_0$ is the intercept, and $b_1$, $b_2$, …, $b_n$ are the coefficients associated with covariates ($X_i$ where $i = 1$ to $n$; $n$ is the number of covariates). The exponentiated coefficient obtained from the logistic part of the model yields OR and provides the effect of the corresponding covariate on excess zeros. Since OR is symmetrical, the inverse of OR provides the effect of the corresponding covariate on the presence of detected polyps compared to no excess zeros.

Observed count component: The PR, NB, or GP models the count of detected polyps after adjusting for an excess amount of no polyps, and it can be expressed as (*Ismail & Jemain, 2007*):

$$\log(E(\text{number of detected polyps} \mid \text{no excess zero})) = \beta_0 + \beta_1 X_1 + \beta_2 X_2 + \ldots + \beta_n X_n$$

where, $\beta_0$ is the intercept, and $\beta_1$, $\beta_2$, …, $\beta_n$ are the coefficients associated with covariates ($X_i$ where $i = 1$ to $n$; $n$ is the number of covariates). The exponentiated coefficient obtained from the count part of the model yields RR. Depending on which count model is used to describe observed counts, the inflated model is referred to as ZIP, ZIGP, or ZINB model. One of the assumptions for the Poisson model is equidispersion, *i.e.*, the mean is equal to the variance. However, sometimes, the count part may have an under- or overdispersion problem even after accounting for excess zeros. This problem can be handled by using a robust variance estimation approach using the Huber-Sandwich estimator method in the ZIP model known as ZIRP (*Hall & Shen, 2010*). In case of the presence of overdispersion in addition to excess zeros, the ZINB or ZIGP model is preferred over the ZIP model (*Ismail & Jemain, 2007*).

### Zero hurdle models

An alternative to zero-inflated models, zero hurdle models can also be used to analyze polyp response data. A zero hurdle model is also a two-part model in which non-zero counts are modeled with truncated count distributions while zero *vs*. non-zero counts are typically modeled with an LR (*Dwivedi et al., 2010*). Although zero-inflated and hurdle models often produce a similar predictive performance, the choice of the model between zero-inflated *vs*. hurdle models depends on the type of zeros (*Dwivedi et al., 2014b*). If zeros are purely the absence of a condition known as true or structural zeros, then zero hurdle models are preferred over zero-inflated models (*Blasco-Moreno et al., 2019*). However, if there are at-risk zeros, false zeros, or sampling or random zeros in addition to structural zeros, then inflated models are recommended over the hurdle models (*Blasco-Moreno et al., 2019*; *Dwivedi et al., 2010*; *Feng, 2021*). The general zero hurdle model can be expressed as:

$$P(Y = y_i) = \begin{cases} f(y_i) & \text{if } y_i = 0 \\ (1 - p_i) * f(y_i) & \text{if } y_i > 0 \end{cases}$$

where $f(y_i)$ can be described with Poisson, NB, or GP distributions, and $p_i$ can be described with logit or probit models for the $i^{th}$ subject. Typically, the LR is used to estimate the probability of observing non-zero counts (*i.e.*, the presence of polyps):

$$\log (\text{odds of presence of polyps}) = b_0 + b_1X_1 + b_2X_2 + \ldots + b_nX_n$$

where, $b_0$ is the intercept, and $b_1$, $b_2$, …, $b_n$ are the coefficients associated with covariates ($X_i$ where i = 1 to n; n is the number of covariates). The exponentiated coefficient obtained from the logistic part of the model yields OR. However, the count part is described using a zero-truncated distribution and provides the likelihood of observing the extent of detected polyps given polyps, and it can be expressed as:

$$\log(E(\text{number of detected polyps} \mid \text{polyps})) = \beta_0 + \beta_1X_1 + \beta_2X_2 + \ldots + \beta_nX_n$$

where, $\beta_0$ is the intercept, and $\beta_1$, $\beta_2$, …, $\beta_n$ are the coefficients associated with covariates ($X_i$ where i = 1 to n; n is the number of covariates). The exponentiated coefficient obtained from the count part of the model yields RR. Depending on the overdispersion beyond zeros, ZHP or ZHNB can be applied to analyze polyp data.

## Statistical analysis

We followed the steps for selecting an appropriate model as specified in Fig. 1. Accordingly, we assessed (a) the presence of excess zeros using a score test (*van den Broek, 1995*) (b) the type of excess zeros using a data generation process (*Dwivedi et al., 2014b*), (c) the presence of overdispersion after removing excess zeros using an overdispersion test (*Cameron & Trivedi, 1990*), and (d) selection of the best-fit model using model fit criteria (*Fernandez & Vatcheva, 2022*). Although we followed Fig. 1 for comparing the performance of different models, this step-by-step approach should not be used for prespecifying the primary models in the statistical analysis, but may be used for validating

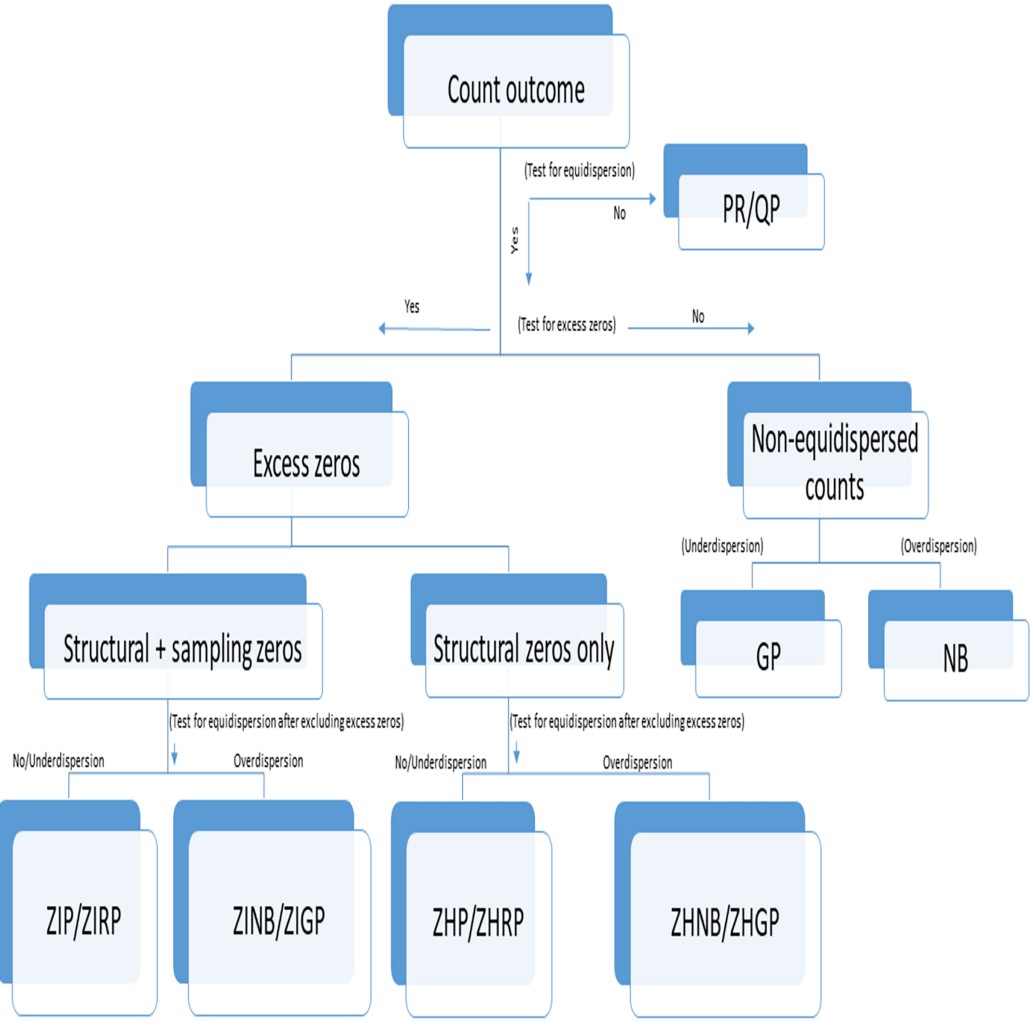

**Figure 1 Flowchart of selection of an appropriate count model in the presence or absence of under- or overdispersion.** PR, Poisson Regression; RP, Robust Poisson; QP, Quasi-Poisson; GP, Generalized Poisson; NB, Negative Binomial; ZIP, Zero-inflated Poisson; ZIRP, Zero-inflated Robust Poisson; ZINB, Zero-inflated Negative Binomial; ZIGP, Zero-Inflated Generalized Poisson; ZHP, Zero Hurdle Poisson; ZHRP, Zero Hurdle Robust Poisson; ZHNB, Zero Hurdle Negative Binomial; ZHGP, Zero Hurdle Generalized Poisson. Structural zeros are true zeros in the absence of a condition while sampling zeros are false zeros likely to be observed due to sampling or methodological or procedural issues or errors.

the robustness of findings from different models in sensitivity analyses. The absence of polyps explained by the LR model was considered as no detected polyps, while no detected polyps explained by the count models are considered as undetected polyps. The distribution of polyps was described using the frequency and proportion as well as the average number of polyps per patient with standard deviation (SD) between groups for each dataset. We applied the standard LR, PR, RP, NB, ZIP, ZIRP, ZINB, ZIGP, ZHP, and ZHNB models in all three datasets to compare the presence of detected polyps and the number of detected polyps between (a) CAC and SC procedures in the example dataset 1, (b) segmental and non-segmental protocols in the example dataset 2, and (c) presence and

absence of a fellowship program in dataset 3. We compared PDR using the same estimand, by computing marginal effect sizes, including the marginal difference in PDR and the marginal odds ratio of PDR (*Long & Jeremy, 2014*). Since a bias-corrected Voung test (*Desmarais & Harden, 2013*) is not appropriate for testing the presence of excess zeros, we primarily used a score test (*van den Broek, 1995*) followed by a bias-corrected Voung test to determine the presence of excess zeros. A significant *p*-value of the score test and bias-corrected Vuong test indicates the presence of an excess amount of no polyps. We applied an overdispersion test (*Cameron & Trivedi, 1990*) and an overdispersion parameter test in the NB regression (*Molla, Muniswamy & Hines, 2012*) after removing excess zeros predicted by inflated models to evaluate if there was the presence of an over- or equi-, or underdispersion issue beyond excess zeros. In the presence of excess zeros in all three datasets, the final inflated model was selected based on model fit indices, including the goodness of fit of the model and information criteria. We summarized the performance of all count models using model fit indices such as the Akaike Information Criterion (AIC), Bayesian Information Criterion (BIC), and negative log likelihood (LL). Lower values of AIC and BIC indicate a better fit of the model, while higher negative LL values indicate a better fit among comparative models. We used the chi-square goodness of fit test to assess differences in observed counts *vs.* expected counts. A non-significant *p*-value of the goodness of fit test indicates a good fit of the model. Among all fit indices, BIC was primarily used to examine the relative fit of the model across all alternative models. Although we used multiple criteria to provide reasons for using a specific model and the most appropriate model describing polyp data in three illustrative studies, the primary models for analyzing data must be specified *a priori* through the study registration or statistical analysis plan (SAP) registration, particularly in interventional studies based on theoretical and simulation considerations. The series of tests and model fit indices used in the study are useful for evaluating assumptions required in the application of primary models and conducting additional sensitivity analyses to validate the findings obtained from primary analyses. The purpose of reporting multiple model fit indices and tests to evaluate assumptions was to validate the robustness of the findings, not to create selection bias issues.

In the primary analysis of randomized studies, the models were adjusted for age, sex, and total procedure time and validated by unadjusted analysis (without any covariate adjustment). In the analysis of the observational study, *a priori*, all the critical covariates, including age, sex, BMI, time of the procedure, and sedation method, were adjusted in the models. Some adjusted variables were removed from the analyses if models, especially ZINB or ZIGP models, did not converge or produce reasonable coefficients. Owing to limited counts of polyps beyond six, we merged six or higher polyp counts in the primary analysis and validated them by considering the actual polyp counts in each dataset. The results of the example dataset 3 were also validated after imputing missing data using the multivariate normal method (*Sullivan et al., 2017*). In primary analyses, we followed statistical analysis guidelines such as evaluation of effect modification and non-linearity before finalizing models as recommended (*Dwivedi, 2022*).

The usefulness of inflated or hurdle models was also determined using simulation studies. The purpose of these simulations was to demonstrate how each statistical method might perform using realistic data but differing from the example datasets used in the study. We conducted three simulation studies (a) using the parameters meeting the criteria of the example dataset, (b) considering the least proportion of zeros (no detected polyps) and no presence of sampling zeros, and (c) considering misspecification of the distributional assumption for each dataset example. For each simulation condition, we generated two random variables, including one binary variable following the binomial distribution with a specified probability and the other variable following a count distribution following either Poisson or NB distribution. A product of these two variables yields a zero-inflated count distribution. In the simulation studies, for example, datasets 1 and 2, we did not include a confounder considering a randomized design, while we introduced a continuous normally distributed confounder for example dataset 3, considering the observational study design. We followed simulation experiments using our previous studies (*Dwivedi et al., 2014a*, *2014b*) and computed average bias (difference between estimated coefficient and true coefficient), relative bias (difference between estimated coefficient and true coefficient divided by the true coefficient), 95% confidence width measuring precision in the estimates, and coverage probability of including the true value in the 95% confidence interval on 1,000 replications. The results of LR were summarized with OR, and the results of count models were summarized with RR along with corresponding 95% confidence interval (CI) and two-sided *p*-value. We also showed the factors associated with detected polyps and the amount of detected polyps obtained using ZIRP and ZINB analyses compared to LR and PR models for the example dataset 3. The estimated PDR from each model was summarized for the primary models. A *p*-value less than 0.05 was considered statistically significant, and the Stata v17.0 (StataCorp, Lakeway Drive, College Station, TX, USA) package was used for all the statistical analyses.

## RESULTS

### Baseline characteristics

The distribution of patient characteristics was found to be similar across three datasets except for preparation quality and procedure time (Table S1). The distribution of polyps was similar in example datasets 1 and 3 compared to dataset 2 (Fig. S1).

### Comparison of CAC *vs.* SC in example dataset 1

In the randomized CAC *vs.* SC study, 87 (41.2%) patients had a polyp in the CAC group while 92 (43%) patients had a polyp in the SC group (Table 1). Testing of excess zeros and overdispersion (Table 2) supported the use of inflated models, particularly ZIP, ZIRP, or ZIGP for analyzing this dataset.

The LR model showed no association between CAC and PDR (OR = 0.90; 95% CI [0.63–1.37]). Similarly, the LR of the ZIP model also yielded no association between CAC and PDR (OR = 0.53; 95% CI [0.23–1.20]). Furthermore, the standard PR (RR = 1.14; 95% CI [0.93–1.41], *p* = 0.215), RP (RR = 1.14; 95% CI [0.89–1.47], *p* = 0.304), and NB (RR = 1.16; 95% CI [0.89–1.51], *p* = 0.265) models yielded no significant association

**Table 1 Distribution of colonoscopy polyps by group in each example dataset.**

| Data | Group | Yes, polyps, $N$ (%) | No polyps, $N$ (%) | Mean (V) | Test for AOD, $p$-value |
|------|-------|------------------|------------------|----------|------------------------|
| Example data 1 | Cap-assisted colonoscopy ($N$ = 211) | 87 (41.23) | 124 (58.77) | 0.92 (2.10) | <0.001 |
| | Standard colonoscopy ($N$ = 214) | 92 (42.99) | 122 (57.01) | 0.76 (1.42) | |
| Example data 2 | Segmental withdrawal protocol ($N$ = 153) | 87 (56.86) | 66 (43.14) | 1.23 (2.43) | <0.001 |
| | Non-segmental withdrawal protocol ($N$ = 158) | 76 (48.10) | 82 (51.90) | 1.05 (1.98) | |
| Example data 3 | After the GI fellowship program ($N$ = 1,035) | 506 (48.89) | 529 (51.11) | 1.01 (2.05) | <0.001 |
| | Before the GI fellowship program ($N$ = 901) | 329 (36.51) | 572 (63.49) | 0.63 (1.09) | |

Note:
V, Variance; AOD, Adjusted overdispersion.

**Table 2 Selection criteria for an appropriate statistical method in each dataset.**

| Data | Bias-corrected Vuong test for the presence of excess zeros* Z-value, $p$-value | Score test for the presence of excess zeros Z-value, $p$-value | Adjusted equidispersion test after removing excess zeros Mean (variance), $p$-value | Adjusted overdispersion test after removing excess zeros $\alpha$, $p$-value |
|------|------|------|------|------|
| Example data 1 | 2.59, $p$ = 0.0048 | 104.85, $p$ < 0.001 | 1.35 (2.14), $p$ = 0.193 | 0.09, $p$ = 0.088 |
| Example data 2 | 2.52, $p$ = 0.0058 | 74.54, $p$ < 0.001 | 1.65 (2.36), $p$ = 0.122 | 0.11, $p$ = 0.049 |
| Example data 3 | 7.41, $p$ < 0.0001 | 392.61, $p$ < 0.001 | 1.19 (1.92), $p$ < 0.001 | 103.06, $p$ < 0.001 |

Note:
* Vuong test with AIC (Akaike Information Criteria) correction; α: overdispersion parameter in the negative binomial model.

between CAC use and a higher number of detected polyps. In contrast, the ZIP model (RR = 1.38; 95% CI [1.05–1.81], $p$ = 0.019) demonstrated a higher number of detected polyps in the CAC group compared to the SC group. These findings were further supported in all inflated or hurdle models, including ZIRP, ZINB, ZIGP, ZHP, and ZHNB analyses (Table 3). These findings were further confirmed through the standardization of estimates. The marginal difference in PDR and the marginal OR of PDR were not found to be significant across models. However, precise estimates were obtained using zero-inflated models (Table 4). These findings were unchanged even without adjusting for any covariates or using all polyp counts (Tables S2 and S3).

The ZIP model yielded higher LL, and lower AIC and BIC values, attesting to its best fit compared to alternative models. In addition, the goodness of fit test only supported the ZIP/ZIRP model ($p$ = 0.11) over other alternative models (Table 5). Based on all the inflated models, it is clear that CAC had a higher number of detected polyps compared to the SC, despite no differences in the PDR.

## Comparison of segmental vs. standard withdrawal time protocols in the example dataset 2

The observed percentage of detected polyps (56.9% vs. 48.1%) was slightly higher in the segmental protocol compared to the standard withdrawal time protocol (Table 1). The average number of polyps was comparable between groups (Table 1). The tests for excess

**Table 3 Comparison of polyp detection rate and the number of detected polyps between groups.**

| Models | Polyp detection rate | | Number of detected polyps | | Estimated PDR |
|---|---|---|---|---|---|
| | OR (95% CI) | *p*-value | RR (95% CI) | *p*-value | |
| **Example dataset 1: CAC *vs*. SC procedures*** | | | | | |
| LR | 0.90 [0.59–1.37] | 0.611 | *NA* | | 57.9% |
| PR | *NA* | | 1.14 [0.93–1.41] | 0.215 | 50.3% |
| RP | *NA* | | 1.14 [0.89–1.47] | 0.304 | 50.3% |
| NB | *NA* | | 1.16 [0.89–1.51] | 0.265 | 56.1% |
| ZIP | 0.53 [0.23–1.20] | 0.127 | 1.38 [1.05–1.81] | **0.019** | 57.9% |
| ZIRP | 0.53 [0.22–1.29] | 0.159 | 1.38 [1.03–1.86] | **0.033** | 57.9% |
| ZINB | 0.51 [0.21–1.24] | 0.138 | 1.39 [1.05–1.84] | **0.021** | 57.9% |
| ZIGP | 0.50 [0.20–1.27] | 0.146 | 1.38 [1.04–1.83] | **0.024** | 57.4% |
| ZHP | 0.90 [0.59–1.37] | 0.611 | 1.40 [1.06–1.85] | **0.018** | 57.9% |
| ZHNB | 0.90 [0.59–1.37] | 0.611 | 1.41 [1.05–1.90] | **0.022** | 57.9% |
| **Example dataset 2: Segmental *vs*. non-segmental withdrawal time protocols*** | | | | | |
| LR | 1.64 [1.00–2.68] | 0.049 | *NA* | | 47.6% |
| PR | *NA* | | 1.09 [0.88–1.35] | 0.430 | 37.6% |
| RP | *NA* | | 1.09 [0.84–1.41] | 0.514 | 37.6% |
| NB | *NA* | | 1.16 [0.89–1.52] | 0.273 | 42.3% |
| ZIP | 3.16 [0.99–10.13] | 0.053 | 0.91 [0.70–1.18] | 0.469 | 47.0% |
| ZIRP | 3.16 [0.67–14.83] | 0.145 | 0.91 [0.67–1.22] | 0.522 | 47.0% |
| ZINB | 15.72 [0.59–419.33] | 0.100 | 0.92 [0.70–1.21] | 0.554 | 46.2% |
| ZIGP | 35.03 [0.89–1,380.39] | 0.058 | 0.89 [0.68–1.17] | 0.404 | 42.2% |
| ZHP | 1.64 [1.00–2.70] | 0.049 | 0.94 [0.72–1.23] | 0.658 | 47.6% |
| ZHNB | 1.64 [1.00–2.70] | 0.049 | 0.95 [0.71–1.26] | 0.701 | 47.6% |
| **Example dataset 3: After the GI fellowship program *vs*. before the GI fellowship program$** | | | | | |
| LR | 1.64 [1.36–1.97] | <0.001 | *NA* | | 56.9% |
| PR | *NA* | | 1.57 [1.42–1.74] | **<0.001** | 45.6% |
| RP | *NA* | | 1.57 [1.37–1.80] | **<0.001** | 45.6% |
| NB | *NA* | | 1.57 [1.36–1.80] | **<0.001** | 45.6% |
| ZIP | 1.52 [1.12–2.05] | **0.006** | 1.31 [1.14–1.51] | **<0.001** | 56.6% |
| ZIRP | 1.52 [1.10–2.08] | **0.01** | 1.31 [1.11–1.55] | **0.001** | 56.6% |
| ZINB | 2.77 [0.99–7.73] | 0.052 | 1.37 [1.14–1.64] | **0.001** | 56.7% |
| ZIGP | 2.11 [0.87–5.14] | 0.101 | 1.31 [1.05–1.62] | **0.015** | 49.3% |
| ZHP | 1.64 [1.36–1.97] | **<0.001** | 1.36 [1.18–1.57] | **<0.001** | 56.9% |
| ZHNB | 1.64 [1.36–1.97] | **<0.001** | 1.44 [1.19–1.76] | **<0.001** | 56.9% |

Notes:
OR, Odds Ratio; RR, Risk Ratio; CI, Confidence Interval; NA, Not Applicable; LR, Logistic Regression; PR, Poisson Regression; RP, Robust Poisson; NB, Negative Binomial; ZIP, Zero-inflated Poisson; ZIRP, Zero-inflated Robust Poisson; ZINB, Zero-inflated Negative Binomial; ZIGP, Zero-inflated Generalized Poisson; ZHP, Zero hurdle Poisson; ZHNB, Zero hurdle Negative Binomial; CAC, Cap-Assisted Colonoscopy; SC, Standard Colonoscopy; GI, Gastroenterology.
* Models were adjusted for age, sex, and total procedure time.
$ Models were adjusted for age, sex, body mass index, time of the day (am *vs*. pm), and sedation type while body mass index was only adjusted in the inflated part of the models. Highlighted *p*-values are significant findings.

**Table 4 Comparison of the polyp detection rate between groups after standardization of estimates.**

| Models | Marginal difference | 95% CI | | Marginal odds ratio | 95% CI | |
|---|---|---|---|---|---|---|
| **Example dataset 1: CAC *vs.* SC procedures**[*] | | | | | | |
| LR | −0.02 | −0.11 | 0.06 | 0.91 | 0.59 | 1.23 |
| PR | 0.04 | −0.02 | 0.11 | 1.18 | 0.87 | 1.49 |
| RP | 0.04 | −0.03 | 0.11 | 1.17 | 0.85 | 1.49 |
| NB | −0.02 | −0.10 | 0.07 | 0.93 | 0.61 | 1.26 |
| ZIP | −0.02 | −0.10 | 0.07 | 0.94 | 0.61 | 1.26 |
| ZIGP | −0.02 | −0.10 | 0.07 | 0.93 | 0.73 | 1.36 |
| **Example dataset 2: Segmental *vs.* non-segmental withdrawal time protocols**[*] | | | | | | |
| LR | 0.10 | 0.00 | 0.21 | 1.52 | 0.89 | 2.15 |
| PR | 0.03 | −0.04 | 0.10 | 1.13 | 0.79 | 1.47 |
| RP | 0.04 | −0.03 | 0.12 | 1.18 | 0.83 | 1.54 |
| NB | 0.12 | 0.02 | 0.23 | 1.65 | 0.96 | 2.34 |
| ZIP | 0.14 | 0.06 | 0.23 | 1.78 | 1.18 | 2.39 |
| ZIGP | 0.15 | 0.02 | 0.23 | 2.49 | 1.05 | 353.50 |
| **Example dataset 3: After the GI fellowship program *vs.* before the GI fellowship program**[$] | | | | | | |
| LR | 0.12 | 0.07 | 0.16 | 1.62 | 1.33 | 1.91 |
| PR | 0.15 | 0.12 | 0.19 | 1.87 | 1.61 | 2.14 |
| RP | 0.11 | 0.07 | 0.14 | 1.55 | 1.34 | 1.76 |
| NB | 0.12 | 0.07 | 0.16 | 1.63 | 1.33 | 1.92 |
| ZIP | 0.12 | 0.08 | 0.16 | 1.62 | 1.34 | 1.90 |
| ZIGP | 0.13 | 0.06 | 0.18 | 1.36 | 1.22 | 1.55 |

**Notes:**
CI, Confidence Interval; LR, Logistic Regression; PR, Poisson Regression; NB, Negative Binomial; ZIP, Zero-inflated Poisson; ZINB, Zero-inflated Negative Binomial; ZIGP, Zero-inflated Generalized Poisson; CAC, Cap-Assisted Colonoscopy; SC, Standard Colonoscopy; GI, Gastroenterology.
[*] Models were adjusted for age, sex, and total procedure time.
[$] Models were adjusted for age, sex, body mass index, time of the day (am *vs.* pm), and sedation type while body mass index was only adjusted in the inflated part of the models.

**Table 5 Comparison of the performance of developed models for evaluating colon polyps.**

| Models | Dataset-1 | | | |
|---|---|---|---|---|
| | AIC | BIC | LL | chi2-GOF (*p*-value) |
| PR/RP | 998.59 | 1,018.85 | −494.30 | 32.1, *p* < 0.001 |
| NB | 975.87 | 1,000.18 | −481.93 | 19.37, *p* < 0.001 |
| ZIP/ZIRP | **962.13** | **1,002.65** | **−471.06** | **7.54, *p* = 0.11** |
| ZINB | 964.05 | 1,008.62 | −471.03 | 10.85, *p* = 0.03 |
| ZIGP | 964.04 | 1,008.61 | −471.02 | 9.45, *p* = 0.09 |
| ZHP | 962.77 | 1,003.29 | −471.38 | 11.41, *p* = 0.04 |
| ZHNB | 964.68 | 1,009.25 | −471.34 | 10.62, *p* = 0.06 |
| | Dataset-2 | | | |
| PR/RP | 878.54 | 897.21 | −434.27 | 31.91, *p* < 0.001 |
| NB | 854.93 | 877.33 | −421.47 | 15.76, *p* < 0.001 |

(Continued)

| Table 5 (continued) | | | | |
|---|---|---|---|---|
| Models | Dataset-1 | | | |
| | AIC | BIC | LL | chi2-GOF (*p*-value) |
| ZIP/ZIRP | **844.10** | **881.43** | **−412.05** | **6.24, *p* = 0.18** |
| ZINB | 842.96 | 884.03 | −410.48 | 6.89, *p* = 0.14 |
| ZIGP | 841.76 | 882.83 | −409.88 | 4.70, *p* = 0.45 |
| ZHP | 846.17 | 883.51 | −413.09 | 7.51, *p* = 0.19 |
| ZHNB | 847.61 | 888.68 | −412.80 | 5.67, *p* = 0.34 |
| | Dataset-3 | | | |
| PR/RP | 5,132.62 | 5,171.60 | −2,559.31 | 222.11, *p* < 0.001 |
| NB | 4,777.87 | 4,822.41 | −2,380.93 | 29.36, *p* < 0.001 |
| ZIP/ZIRP | 4,846.50 | 4,896.61 | −2,414.25 | 68.58, *p* < 0.001 |
| ZINB | **4,776.62** | **4,832.30** | **−2,378.31** | **9.06, *p* = 0.06** |
| ZIGP | 4,777.69 | 4,833.38 | −2,378.85 | 108.56, *p* < 0.001 |
| ZHP | 4,850.69 | 4,917.51 | −2,413.34 | 91.45, *p* < 0.001 |
| ZHNB | 4,785.22 | 4,868.78 | −2,377.63 | 19.51, *p* = 0.002 |

**Note:**
PR, Poisson Regression; RP, Robust Poisson; NB, Negative Binomial; ZIP, Zero-inflated Poisson; ZIRP, Zero-inflated Robust Poisson; ZINB, Zero-inflated Negative Binomial; ZIGP, Zero-inflated Generalized Poisson; ZHP, Zero hurdle Poisson; ZHNB, Zero hurdle Negative Binomial; AIC, Akaike Information Criterion; BIC, Bayesian Information Criterion; LL, Log-likelihood; GOF, Goodness of Fit. Highlighted data indicates the best fit of the model.

zeros and overdispersion supported the use of ZIP or ZIRP models for analyzing this dataset (Table 2).

The odds of detecting polyps were associated (OR = 1.64; 95% CI [1.00–2.68], *p* = 0.049) with segmental protocol in the LR model. Although a slightly higher PDR was noticed with the segmental protocol compared to the standard protocol, the difference did not attain a statistically significant level in ZIP (OR = 3.16; 95% CI [0.99–10.13], *p* = 0.053) and ZIRP (OR = 3.16; 95% CI [0.67–14.83], *p* = 0.145) models. Segmental protocol did not yield significantly higher polyp counts compared to standard protocol, as confirmed by standard count, inflated, and hurdle models (Table 3). After standardization, overall PDR was significantly higher in the segmental protocol compared to the standard protocol (Table 4). These findings were further supported by unadjusted models or analysis of all polyp counts (Tables S2 and S3).

As indicated by excess zeros and overdispersion tests, all the inflated and hurdle models yielded a better fit compared to PR or NB models, as confirmed by lower AIC and BIC values and higher negative LL values (Table 5). Although ZIGP or ZINB showed slightly improved fit compared to the ZIP model, ZIGP and ZINB produced unusually high effect sizes for PDR, especially in the absence of overdispersion, indicating that ZIRP should be preferred for this dataset.

## Effect of implementation of fellowship program on colon polyps in the example dataset 3

We observed increased PDR (48.9% *vs.* 36.5%) with increased polyp counts (1.0 *vs.* 0.63) after the introduction of the GI fellowship program compared to the years without the GI

fellowship program (Table 1). We observed a significantly higher amount of no polyps as identified by score ($p < 0.001$) and bias-corrected Vuong ($p < 0.001$) tests (Table 2). However, the overdispersion was also noticed with or without removing excess zeros ($p < 0.001$), indicating the use of ZINB or ZIGP models (Tables 1 and 2).

In the adjusted analysis, LR (OR = 1.64; 95% CI [1.36–1.97], $p < 0.001$) confirmed higher PDR with the GI fellowship program compared to the years without the fellowship program. However, the standard count models, including PR (RR = 1.57; 95% CI [1.42–1.74], $p < 0.001$), RP (RR = 1.57; 95% CI [1.36–1.80], $p < 0.001$), NB (RR = 1.57; 95% CI [1.36–1.80], $p < 0.001$), and ZIRP (RR = 1.31; 95% CI [1.11–1.55], $p = 0.001$) also showed a strong association between the number of detected polyps and the GI fellowship program (Table 3). Finally, the ZINB model demonstrated that the GI fellowship period was associated with polyp counts only (RR = 1.37; 95% CI [1.14–1.64], $p = 0.001$). All models showed a significantly higher PDR associated with the GI fellowship program, as confirmed by the marginal difference in PDR or marginal OR of PDR (Table 4). These findings remained unchanged with ZIGP (Table 3), fully adjusted models, or analysis of all polyp counts (Tables S2 and S3). The association of the GI fellowship program with PDR and polyp counts was more pronounced after imputing missing data, indicating the importance of accounting for potentially missing at random data (Table S4).

The model fit characteristics favored ZINB over other alternative models by yielding lower AIC or BIC values and non-significant goodness of fit of the model ($p = 0.06$) (Table 5). The two-part analysis with the ZINB model suggested that male sex (RR = 1.51; 95% CI [1.32–1.74], $p < 0.001$), older age per year (RR = 1.02; 95% CI [1.01–1.03], $p = 0.001$), time of the procedure after midday (RR = 1.36; 95% CI [1.17–1.58], $p < 0.001$) and moderate sedation (RR = 0.60; 95% CI [0.39–0.91], $p = 0.018$) were associated with the number of detected polyps but not with a presence of detected polyps. In contrast, only increases in BMI ($Kg/m^2$) were associated with PDR (OR = 1.14; 95% CI [1.04–1.26], $p = 0.008$) (Table 6). Some of these factors remained significant in fully adjusted models as well as ZIRP analysis (Table S5).

## Simulation studies

All the inflated and hurdle models produced reliable estimates with high coverage probability in simulation studies corresponding to dataset 1, involving mixed zeros or only structural zeros with a low proportion of zeros. However, ZINB followed by ZHNB and ZIRP yielded reliable estimates with high coverage if the data generation process followed an NB distribution with sampling zeros instead of a Poisson distribution. Simulation studies corresponding to dataset 2 favored the ZIRP model over other models in all scenarios. For dataset 3, simulation studies favored ZIRP over other models regardless of the types of zeros and the number of zeros if the data followed the Poisson distribution. However, ZHNB produced a better fit over other models if count data followed an NB distribution with limited sampling zeros. In the case of sampling zeros, like in our example datasets, ZHNB produced less precise estimates compared to ZINB with comparable coverage and bias. Considering the instability in parameter estimates and convergence issues with ZINB and ZIGP in some cases, ZIRP should be preferred in most cases since

**Table 6 Associated factors with polyp detection rate and the number of detected polyps in example data 3.**

| Models | LR | | ZIRP* | | ZINB* | |
|---|---|---|---|---|---|---|
| **Polyp detection rate (PDR)** | OR (95% CI) | *p*-value | OR (95% CI) | *p*-value | OR (95% CI) | *p*-value |
| Fellowship-presence | 1.64 [1.36–1.97] | <0.001 | 1.52 [1.10–2.08] | 0.01 | 2.77 [0.99–7.73] | 0.052 |
| Body mass index (kg/m$^2$) | 1.02 [1.01–1.04] | 0.003 | 1.03 [1.01–1.06] | 0.02 | 1.14 [1.04–1.26] | 0.008 |
| Age (years) | 1.02 [1.01–1.04] | 0.001 | | | | |
| Sex-male | 1.55 [1.28–1.89] | <0.001 | | | | |
| Time of day-PM | 1.48 [1.21–1.80] | <0.001 | | | | |
| Sedation-moderate | 0.75 [0.39–1.44] | 0.395 | | | | |
| | **PR** | | **RP** | | **NB** | |
| **Detected polyp counts** | RR (95% CI) | *p*-value | RR (95% CI) | *p*-value | RR (95% CI) | *p*-value |
| Fellowship-presence | 1.57 [1.42–1.74] | <0.001 | 1.31 [1.11–1.55] | 0.001 | 1.37 [1.14–1.64] | 0.001 |
| Age (years) | 1.02 [1.01–1.03] | <0.001 | 1.01 [1.00–1.02] | 0.005 | 1.02 [1.01–1.03] | 0.001 |
| Body mass index (kg/m$^2$) | 1.02 [1.01–1.03] | <0.001 | | | | |
| Sex-male | 1.53 [1.38–1.69] | <0.001 | 1.44 [1.25–1.65] | <0.001 | 1.51 [1.32–1.74] | <0.001 |
| Time of day-PM | 1.38 [1.24–1.55] | <0.001 | 1.31 [1.11–1.54] | 0.001 | 1.36 [1.17–1.58] | <0.001 |
| Sedation-moderate | 0.61 [0.47–0.79] | <0.001 | 0.61 [0.44–0.84] | 0.002 | 0.60 [0.39–0.91] | 0.018 |

Notes:
OR, Odds Ratio; RR, Risk Ratio; CI, Confidence Interval; LR, Logistic Regression; PR, Poisson Regression; ZIRP, Zero-inflated Robust Poisson; ZINB, Zero-inflated Negative Binomial.
* Age, sex, time of the day, and sedation were removed from the PDR part of the model, while body mass index was removed from the polyp count part of the model.

**Table 7 Simulation studies for evaluating the performance of different models corresponding to each of the three example datasets.**

| | *p* = 60.5%; β = 0.40; CD = Poisson | | | | *p* = 16%; β = 0.40; CD = Poisson | | | | *p* = 66%; β = 0.40; CD = Poisson | | | |
|---|---|---|---|---|---|---|---|---|---|---|---|---|
| *N* = 425 | TZ = 52.5%; SZ = 18% | | | | TZ = 16%; SZ = 1% | | | | TZ = 52.6%; SZ = 28.9%; alpha = 0.50 | | | |
| | Bias | RB | CW | CP | Bias | RB | CW | CP | Bias | RB | CW | CP |
| PR | 0.207 | 0.517 | 0.432 | 0.538 | 0.059 | 0.147 | 0.180 | 0.697 | 0.213 | 0.533 | 0.434 | 0.507 |
| RP | 0.207 | 0.517 | 0.593 | 0.737 | 0.059 | 0.147 | 0.244 | 0.846 | 0.213 | 0.533 | 0.709 | 0.809 |
| NB | 0.207 | 0.517 | 0.642 | 0.782 | 0.059 | 0.147 | 0.273 | 0.886 | 0.213 | 0.533 | 0.765 | 0.838 |
| ZIP | 0.004 | 0.009 | 0.562 | 0.954 | −0.002 | −0.004 | 0.183 | 0.947 | −0.047 | −0.118 | 0.527 | 0.883 |
| ZIRP | **0.004** | **0.009** | **0.560** | **0.951** | **−0.002** | **−0.004** | **0.183** | **0.946** | **−0.047** | **−0.118** | **0.651** | **0.942** |
| ZINB | 0.008 | 0.019 | 0.560 | 0.951 | −0.001 | −0.003 | 0.183 | 0.946 | **0.000** | **0.000** | **0.651** | **0.942** |
| ZIGP | 0.003 | 0.008 | 0.576 | 0.953 | −0.002 | −0.004 | 0.182 | 0.946 | 0.070 | 0.174 | 0.982 | 0.970 |
| ZHP | 0.004 | 0.009 | 0.562 | 0.954 | −0.002 | −0.004 | 0.183 | 0.947 | −0.047 | −0.118 | 0.527 | 0.883 |
| ZHNB | 0.008 | 0.019 | 0.578 | 0.958 | −0.001 | −0.003 | 0.185 | 0.949 | **0.000** | **0.000** | **0.743** | **0.952** |
| | *p* = 50.6%; β = 0.20; CD = Poisson | | | | *p* = 16.6%; β = 0.20; CD = Poisson | | | | *p* = 48.2%; β = 0.20; CD = Poisson | | | |
| *N* = 311 | TZ = 48%; SZ = 5% | | | | TZ = 16%; SZ = 1% | | | | TZ = 48.1%; SZ = 1%; alpha = 0.10 | | | |
| PR | 0.165 | 0.827 | 0.363 | 0.531 | 0.053 | 0.264 | 0.220 | 0.778 | 0.169 | 0.847 | 0.220 | 0.300 |
| RP | 0.165 | 0.827 | 0.565 | 0.816 | 0.053 | 0.264 | 0.293 | 0.907 | 0.169 | 0.847 | 0.525 | 0.772 |
| NB | 0.165 | 0.827 | 0.673 | 0.899 | 0.053 | 0.264 | 0.324 | 0.924 | 0.169 | 0.847 | 0.791 | 0.950 |
| ZIP | 0.000 | 0.002 | 0.398 | 0.960 | −0.002 | −0.009 | 0.224 | 0.948 | −0.004 | −0.021 | 0.220 | 0.849 |
| ZIRP | **0.000** | **0.002** | **0.395** | **0.955** | **−0.002** | **−0.009** | **0.224** | **0.948** | **−0.004** | **−0.021** | **0.292** | **0.944** |

| Table 7 (continued) | | | | | | | | | | | | |
|---|---|---|---|---|---|---|---|---|---|---|---|---|
| | $p = 50.6\%$; $\beta = 0.20$; CD = Poisson | | | | $p = 16.6\%$; $\beta = 0.20$; CD = Poisson | | | | $p = 48.2\%$; $\beta = 0.20$; CD = Poisson | | | |
| $N = 311$ | TZ = 48%; SZ = 5% | | | | TZ = 16%; SZ = 1% | | | | TZ = 48.1%; SZ = 1%; alpha = 0.10 | | | |
| ZINB | 0.002 | 0.008 | 0.395 | 0.955 | −0.002 | −0.008 | 0.224 | 0.948 | −0.002 | −0.012 | 0.292 | 0.944 |
| ZIGP | 0.000 | 0.001 | 0.395 | 0.957 | −0.002 | −0.010 | 0.223 | 0.947 | −0.007 | −0.034 | 0.295 | 0.950 |
| ZHP | 0.000 | 0.002 | 0.398 | 0.960 | −0.002 | −0.009 | 0.224 | 0.948 | −0.004 | −0.021 | 0.220 | 0.849 |
| ZHNB | 0.002 | 0.008 | 0.407 | 0.963 | −0.002 | −0.008 | 0.228 | 0.952 | −0.002 | −0.012 | 0.295 | 0.945 |
| | $p = 60.2\%$; $\beta = 0.55$; CD = Poisson | | | | $p = 15.6\%$; $\beta = 0.55$; CD = Poisson | | | | $p = 57.7$; $\beta = 0.55$; CD = Poisson | | | |
| $N = 1,936$ | TZ = 56%; SZ = 10.2% | | | | TZ = 15%; SZ = 1% | | | | TZ = 56.1%; SZ = 4%; alpha = 0.30 | | | |
| PR | 0.282 | 0.512 | 0.182 | 0.003 | 0.076 | 0.139 | 0.082 | 0.099 | 0.281 | 0.511 | 0.116 | 0.001 |
| RP | 0.282 | 0.512 | 0.275 | 0.025 | 0.076 | 0.139 | 0.111 | 0.238 | 0.281 | 0.511 | 0.279 | 0.020 |
| NB | 0.282 | 0.513 | 0.315 | 0.039 | 0.076 | 0.139 | 0.124 | 0.317 | 0.282 | 0.512 | 0.370 | 0.075 |
| ZIP | 0.001 | 0.002 | 0.218 | 0.941 | 0.001 | 0.001 | 0.083 | 0.953 | −0.033 | −0.060 | 0.118 | 0.691 |
| ZIRP | **0.001** | **0.002** | **0.218** | **0.943** | **0.001** | **0.001** | **0.083** | **0.951** | −0.033 | −0.060 | 0.185 | 0.900 |
| ZINB | 0.003 | 0.005 | 0.218 | 0.943 | 0.001 | 0.002 | 0.083 | 0.951 | 0.000 | 0.000 | 0.185 | 0.900 |
| ZIGP | 0.001 | 0.002 | 0.223 | 0.940 | 0.001 | 0.001 | 0.083 | 0.951 | −0.024 | −0.044 | 0.212 | 0.936 |
| ZHP | 0.001 | 0.002 | 0.218 | 0.943 | 0.001 | 0.001 | 0.083 | 0.953 | −0.033 | −0.060 | 0.118 | 0.690 |
| ZHNB | 0.003 | 0.005 | 0.221 | 0.944 | 0.001 | 0.002 | 0.084 | 0.954 | **0.000** | **0.000** | **0.198** | **0.962** |

Note:
Bias, average bias; RB, relative bias; CW, confidence width; CP, coverage probability; CD, count distribution; P, the overall proportion of zeros; TZ, true/structural zeros; SZ, sampling/false zeros; alpha, dispersion parameter in negative binomial; CD, count distribution; $\beta$, true regression coefficient; PR, Poisson Regression; RP, Robust Poisson; NB, Negative Binomial; ZIP, Zero-inflated Poisson; ZIRP, Zero-inflated Robust Poisson; ZINB, Zero-inflated Negative Binomial; ZIGP, Zero-inflated Generalized Poisson; ZHP, Zero hurdle Poisson; ZHNB, Zero hurdle Negative Binomial. Bold text indicates best performance for the model.

ZHNB produced less precise 95% CI compared to ZIRP. However, ZHNB may be preferred over ZIRP when the underlying count distribution follows an NB distribution with moderate overdispersion (Table 7).

## DISCUSSION

In our study, we observed a significant effect of the CAC in detecting a higher number of polyps compared to SC, without any difference in PDR between procedures using a ZIP model. However, the standard approaches produced potentially inaccurate findings that CAC was neither associated with higher PDR nor with the number of detected polyps. In another example dataset, both the standard LR and ZIP models produced the same findings that there was slightly higher PDR in the segmental withdrawal time protocol compared to the standard withdrawal time protocol. The application of the ZIP model also suggested an insignificant difference in detected polyp counts between the segmental protocol and the non-segmental protocol. In another data example, our proposed analytic approach using ZIRP or ZINB suggested that the number of detected polyps increased after the introduction of the GI fellowship program. The effect sizes (OR/RR) obtained from the inflated models were different than standard approaches, indicating the standard approaches may produce biased estimates, as confirmed by our simulation studies, even if the conclusions do not change. The standard count models may produce biased estimates in the presence of excess zeros, which is the case in our example datasets. The logistic

regression may produce an inefficient model (because of data dichotomization), but it may also produce biased estimates, especially in the presence of sampling zeros, which are likely to be observed in colonoscopic studies. While the LR suggested most cofactors were associated with detected polyps, the application of the ZIP model suggested that fellowship was the key factor associated with higher PDR. Contrary to the standard count models, the ZINB model produced factors such as male sex, older age, time of the procedure, and sedation type associated with the polyp counts, while only BMI was associated with PDR. These findings clearly support the application of zero-inflated models over the standard approaches. The application of zero-inflated models provides complete information, including the differences in the presence or absence of polyps as well as differences in the extent of detected polyps compared to single distribution models. Moreover, the marginal estimates obtained from zero-inflated models, particularly ZIP models, yielded more precise estimates for all three datasets, indicating that there is a gain in the use of zero-inflated models in the presence of excess zeros. Many colonoscopy screening trials might have reported potentially inadequate conclusions or inappropriate effect sizes due to the inappropriate use of statistical analysis methods. It is known that inappropriate use of methods, not appropriate with evidence-based biostatistics, may lead to false conclusions (*Dwivedi, 2022*; *Dwivedi & Shukla, 2020*).

Although our study clearly demonstrates that there was an excess amount of no polyps in all three datasets indicating that a zero-inflated count model was indeed required to analyze these studies, the simple PR model was sometimes applied to analyze colorectal polyps (*Chan, Cohen & Spiegel, 2009*; *Davies et al., 2023*; *Hillyer et al., 2014*; *Liu et al., 2020*; *Othman et al., 2017*). In all three datasets, more than half of the patients had no polyps. Multiple studies (*Asadzadeh Aghdaei et al., 2017*; *Lee et al., 2013*) showed that PDR ranges between 20% to 59.9%, indicating a disproportionally higher number of no polyps than detected polyps in colonoscopy studies, attesting to the application of zero-inflated models. Although the NB model produced a better or similar fit based on BIC criteria than the inflated models sometimes, the standard count models yield false findings in mixture data distributions (*Campbell, 2021*), which is also validated in our simulation studies. For example, in the CAC study, we observed a lower proportion of detected polyps, while increased polyp counts were associated with CAC compared to the SC using inflated models. However, the NB model considered all no polyps and undetected polyps as part of a single count distribution, indicating no difference in the number of detected polyps. Therefore, even if the NB model fits well for the dataset, it should be avoided in the case of an excess amount of zeros (*Blasco-Moreno et al., 2019*; *Fernandez & Vatcheva, 2022*). Moreover, sometimes the model fit characteristics do not appropriately guide the proper use of the models (*Campbell, 2021*). However, reporting model fit characteristics is required in some study objectives (*Dwivedi, 2022*). Our purpose in reporting multiple model fit characteristics was to assess the robustness of findings using a specific model.

Multiple studies support our findings obtained through ZIP models and refute findings obtained from binary models. In a meta-analysis of 11 studies reporting a comparison of CAC with SC, seven studies did not show any significant improvement in PDR with CAC compared to SC (*Mir et al., 2017*). In contrast, several studies (*Kim et al., 2015*; *Othman*

*et al., 2017*) demonstrated that CAC yielded significantly more polyps than SC, supporting our findings obtained through the ZIP or inflated models, but were missed by the standard PR/NB model. Although increasing withdrawal times were shown to be associated with increased PDR (*Haghbin, Zakirkhodjaev & Aziz, 2023*), the segmental protocol of spending at least 3 min to the right colon increased PDR but did not show a significant benefit over the standard protocol of withdrawal time, as confirmed in our ZIP models for detection of more polyps. Several studies in separate analyses of dichotomized and count forms of colon polyps confirmed our findings obtained through ZIP or ZINB models that increasing fellows' participation is associated with increased PDR (*Qayed et al., 2017*) and the detection of more polyps (*Elhanafi et al., 2017*; *Facciorusso et al., 2020*; *Lee et al., 2011*; *Qayed et al., 2017*).

We observed a significant dispersion issue in the third example dataset, which we could easily address by performing ZIRP analysis. In fact, we suggest using a robust variance estimation approach in ZIP (ZIRP) or standard PR model (RP) regardless of the dispersion issue, as it can estimate the appropriate variance in under- or slightly overdispersion situations (*Campbell, 2021*; *Payne et al., 2018*). If there is no overdispersion and no excess zero issues then one should prefer using QP or GP over the ordinary PR model (*Harris, Yang & Hardin, 2012*). However, if there are no excess zeros but there is an overdispersion then analysts may prefer using the NB model (*Fernandez & Vatcheva, 2022*). If there is a non-equidispersion issue in addition to excess zeros then it is recommended to use ZINB or ZIGP (*Ismail & Jemain, 2007*; *Yang, Hardin & Addy, 2009*). However, these models may produce an overfit issue by yielding unusually high effect sizes if there is an underdispersion in count values, excluding excess zeros (*Blasco-Moreno et al., 2019*; *Campbell, 2021*; *Fernandez & Vatcheva, 2022*) which is the case mostly occurring in colorectal polyp data as we observed in our randomized studies.

We observed unstable coefficients, especially in the analysis of example dataset 2, and observed convergence issues, especially with ZINB and ZIGB for analyzing datasets 1 and 3 without adjusting for covariates. The methodologists need to consider event per variable (EPV) which typically requires 10 EPV for the LR part of the inflated model (*Peduzzi et al., 1996*), avoid using NB or ZINB models if there is no overdispersion beyond excess zeros (*Fernandez & Vatcheva, 2022*), avoid adjusting covariates that are not needed for the inflated part of the model, and consider the appropriate sample size (*Williamson et al., 2022*). In addition, ZHNB may be preferred over ZINB or ZIGP models to avoid convergence issues if there are limited (<10%) sampling zeros. Although we observed similar conclusions with any inflated or hurdle models in both randomized studies, the overdispersion tests showed no significant presence of overdispersion, indicating the use of ZIRP over the ZINB, ZIGP, ZHP, or ZHNB model. We recommend using ZIRP in such a situation, which can estimate robust variance even with a misspecification of the distributional assumption, as shown in different simulation scenarios. However, in the presence of overdispersion along with excess zeros, as we observed in the example dataset 3, we recommend exploring ZINB or ZHNB models along with the ZIRP model. Our simulation studies also confirmed these recommendations. In colonoscopic procedures, there is a chance of having undetected polyps (false or sampling zeros) even if there is an

existing polyp (*Pamudurthy, Lodhia & Konda, 2020*). Therefore, zero-inflated models are the best choice for analyzing colorectal polyps according to the best biostatistics and evidence-based biostatistics practices (*Dwivedi, 2022*; *Dwivedi & Shukla, 2020*). If the analysts are unsure about the type of zeros, require ease of interpretation, or face convergence issues or instability in estimated parameters, particularly with ZINB or ZIGP models, then zero hurdle models may be used over the zero-inflated models. In our study, we did not notice differences in hurdle and inflated models for randomized studies. However, considering less precise estimates with hurdle models compared to inflated models as obtained in simulation studies, inflated models, preferably ZIRP should be used for analyzing polyp data. However, hurdle models may be preferred over inflated models to avoid convergence or instability issues in studies yielding overdispersion beyond excess zeros.

We also facilitated bias, precision, and power comparisons for inflated or hurdle models over single distribution models in the presence of excess zeros. However, we did not reevaluate other published trials in this area to demonstrate further change in the conclusions by using inflated models. Another limitation of our study is missing some confounders in dataset 3 and the application of these models in studies with repeated measures design. The major clinical heterogeneity occurs due to the bowel preparation quality and the experience of the gastroenterologists performing the colonoscopy. In randomized studies, these were balanced due to strict randomization. We adjusted these sources of variation in the analysis of the observational study. Although our goal of the study was to demonstrate the usefulness of inflated/hurdle models for analyzing count data in association and interventional studies, there are emerging machine learning models for handling overdispersion and excess zeros which could be compared with standard inflated/hurdle models in prediction studies (*Ben Khedher & Yun, 2024*; *Haghani, Sedehi & Kheiri, 2017*; *Sidumo, Sonono & Takaidza, 2024*). We highlight that colonoscopy studies should use polyp counts as the quality measure, and methodologists should jointly analyze the PDR with polyp counts using inflated models. To the best of our knowledge, our study for the first time demonstrates that count outcome data generated in colposcopy studies should be analyzed with inflated models as opposed to the standard count or binary LR models. We also demonstrate that ZIRP is more appropriate for underdispersed or equidispersed data, which is typically observed in randomized studies. Our work provides a guidance document for selecting an appropriate choice of count models in view of excess zeros, dispersion, and type of excess zeros, and related statistical procedures for analyzing polyp data. For promoting value-based biostatistics practice, we strongly recommend the use of zero-inflated models, particularly ZIRP for detecting the number of polyps in the primary analysis, regardless of checking the fitting parameters.

## CONCLUSIONS

Studies on colorectal polyps are predominantly analyzed with the LR model, leading to inefficient estimates and potentially inaccurate findings. We observed that colon cancer screening studies yielded an excess amount of no polyps, which were unexplained by the standard count models. We demonstrated that the standard analytic approaches for

analyzing colorectal polyps produced biased effect sizes due to poor model fit characteristics, inaccurate interpretations, and conclusions as compared to the inflated models. ZIP with robust variance estimation was found to be the optimal method for analyzing polyp data in randomized studies, while the ZINB model showed a better fit in an observational study. Considering the data generation process of colonoscopic studies and simulation studies, we strongly endorse the use of zero-inflated models for evaluating colorectal polyps, regardless of fitting characteristics in colonoscopy screening studies, for proper clinical interpretation of data and accurate reporting of findings.

## ACKNOWLEDGEMENTS

We acknowledge the Office of Research resources for providing suitable assistance with the statistical work. We appreciate Vishwajeet Singh's providing preliminary data analysis and feedback. We also appreciate the efforts of investigators and participants who were involved in conducting and maintaining datasets.

### Funding

The authors received no funding for this work.

### Competing Interests

The authors declare that they have no competing interests.

### Author Contributions

- Alok K. Dwivedi conceived and designed the experiments, performed the experiments, analyzed the data, prepared figures and/or tables, authored or reviewed drafts of the article, and approved the final draft.
- Sherif E. Elhanafi conceived and designed the experiments, performed the experiments, authored or reviewed drafts of the article, and approved the final draft.
- Mohamed O. Othman conceived and designed the experiments, performed the experiments, authored or reviewed drafts of the article, and approved the final draft.
- Marc J. Zuckerman conceived and designed the experiments, performed the experiments, authored or reviewed drafts of the article, and approved the final draft.

### Data Availability

The raw data is available in the Supplemental File.

### Supplemental Information

Supplemental information for this article can be found online at http://dx.doi.org/10.7717/peerj.19504#supplemental-information.

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
