# Peer review of "Zero-inflated models for the evaluation of colorectal polyps in colon cancer screening studies—a value-based biostatistics practice"

_PeerJ, doi:10.7717/peerj.19504_

## Round 0.1 · original submission · Major Revisions

Thank you for your interesting submission. We have some very useful comments from two reviewers on your work. They have collectively identified a number of questions for you to address. I’ll add a few more from myself after summarising some of their points below. Addressing all of the suggestions/points may require substantial changes to your work, so I won’t make too many minor/specific comments at this time. If you feel that you can address these points, either by making changes to your manuscript or by rebuttal, I would be very happy to see a new version of your manuscript (with all changes tracked) along with a letter outlining those changes and/or explaining why you don’t believe you need to make changes for each of our reviewers’ and my points.

For Reviewer #1, I think that their point around what is currently being done would be very useful in providing context. While I understand their point about machine learning approaches, I’ll leave it up to you whether you want to expand the focus of your discussion or keep it restricted to the set of techniques you’ve explored. I think there is much merit in their suggestion to look at simulations as well as your data sets as case studies. You could base your simulation parameters on observed aspects of your data sets. All of their comments warrant careful consideration and responses, either through revisions or rebuttals.

Reviewer #2 raises an important point about using the same estimand across data sets. There is a risk that the reader bases their interpretations of the techniques on the (lack of) statistical significance for some models compared to others (as Gelman has said, the difference between statistically significant and not statistically significant isn’t itself statistically significant). Their point about pre-specifying models is also an important one, especially when looking at data from intervention studies, where models (and their covariates) should have been clearly specified in the study registration. All of their comments warrant careful consideration and responses, either through revisions or rebuttals.

It seems plausible that the gains from zero-inflation would depend on, among other things, the patient mix, which could inform the proportion of zeros. This might be an interesting aspect to explore through simulations (decreasing/increasing the rate of zeros from those you observe).

In terms of hurdle models, it seems to me that your main point here is to correctly specify the model without losing useful information. I agree that if there is a mixture of missed polyps and patients without any polyps, zero-inflation makes sense, but leaving this for the discussion (Line 382 onward) seems less helpful to readers than explaining this point earlier (around Line 101). Given that logistic regression asks very different questions (and this might be the right question to ask in some situations), and given the likely issues with Poisson regression, you’re already comparing a range of approaches that readers might consider. So, why not include hurdle models in your results? You suggest that readers could look at these (Line 395 onward) and I’m not clear why this would be left for further study rather than looked at here for completeness.

I can’t entirely agree with your point about relative risks versus odds ratios (Lines 152–155). While I will sometimes use robust standard errors with Poisson regression for ease of communicating findings to an audience less familiar with ORs, there are good arguments in favour of ORs (including, but not limited to, symmetry). In any case, you report ORs for the dichotomized outcome/inflation, so I’m not sure that this text is needed. Note that associations with the actual number of polyps wouldn’t be a relative risk but a risk ratio (see, e.g., Table 3’s notes but also Lines 166–167).

This relates back to a point from Reviewer #2, but I can’t agree that significance tests for excess zeros, etc. is advisable (Line 214 onward, Figure 1) as this presents many of the same problems as testing residuals for normality, etc. I would suggest that theoretical considerations should drive the selection of the appropriate model(s) here, with sensitivity analyses as appropriate. I would expect such an automated approach to bias the type I error rate away from the nominal level and this is something that readers, in my opinion, ought to be concerned with (and something that could come through in simulations).

You discuss looking at both AIC and BIC (Line 235 onward) but these have different goals and examining both risks a contradiction/inconsistency in model selection. Which measure do you feel is better in the context you’re considering? The question is then further complicated by including LLs and GoF tests. The more criteria that are used, the more potentially inconsistent metrics need to be reconciled and I suggest fewer rather than more here.

Line 244: Note that “STATA” should be “Stata”.

You say that you’re not looking at power (Line 398), but this is something that some readers will be concerned about. I would favour precision, but either would be a (relatively straightforward) extension here with simulations. I really like the three data sets as case studies, but taking a simulation approach using parameters informed by these data sets would allow a much richer examination of the questions (bio)statisticians might have when faced with similar data.

Looking at Table 3, there are some clear signs of model instability, e.g. the zero-inflated components for data set 2. There have been many advances on Peduzzi, et al.’s work on EPVs in logistic regression, and I think that readers will want some guidance around this aspect here.

While the case studies are interesting, these raise other challenges around modelling non-linear associations (e.g., with age) and effect modification. How were these assessed/determined here?

Reviewer 1 ·

Basic reporting

no comment

Experimental design

no comment

Validity of the findings

no comment

Additional comments

Did authors explore the prevalence of using zero inflated models in current medical literature? Any preliminary research by authors or adding reference will add value to this manuscript.

Authors should present the baseline characteristics of each dataset in the table 1 (or in the supplementary material as appropriate) and should have brief description of each in the results section.

Authors did not plan to conduct simulations to compare the performance (in terms of relative bias etc) of these models. Appreciate if authors explain reasoning behind their approach in the manuscript.

I did not see discussion on using modern machine learning techniques in this perspective. I think authors should enhance their discussion being more inclusive of different techniques and methods.

How much clinical heterogeneity may have influenced the performance of models presented in this manuscript? A few lines of description will be appreciated.

Authors should consider minimising some unnecessary text from the results section as it is too long to read at the moment.

A number of potential limitations also apply to this manuscript. I suggest discussing these will be really helpful to readers and open avenues for future work in this domain.

Table 1 presents the distribution of number of colon polyps across three data sets. I think minimum and maximum values should also be added in addition to the mean (and variance) for better clarity. I also suggest adding distributional graph for each dataset showing number of colon polyps in the supplementary material for better clarity.

Table 3, is it okay to say OR zero inflated models when reporting the polyp detection rate across the three datasets? When detecting the number of polyps using zero inflated models, similar estimates with directions were observed. Is it safe to conclude the zero inflated models should be used regardless of checking the fitting parameters as clinical researchers hardly follow in practice.

Table 4 presents the comparison of performance measures among various model choices for number of detected polyps. Zero inflated models do not differ too much in their performance measures. Of course, precision is important but trying to understand about any harm if you do not use exact zero inflated model. Can we say zero inflated models are better fit in general?

Table 5, can we include all factors in when fitting ZIRP and ZINB for polyp detection? Reason for not adding should be provided in the table footnote.

Reviewer 2 ·

Basic reporting

Thank you for the opportunity to review this very interesting manuscript. I think this research has great potential and is well worth pursuing. However, there are several issues.

First and foremost, when comparing the same methods, it would be ideal to compare them based on how they all estimate the same quantity (i.e., the same estimand). Can all the models considered estimate the PDR? I believe fitting a ZIP model to the data will still allow one to estimate the “implied PDR” of the fitted model, but this requires some standardization.

Furthermore, the authors cite Campbell but then go on to suggest that the best model is chosen based on a series of goodness of fit tests. Is this the best recommendation or should the primary model be pre-specified to avoid any model selection bias?

Finally with respect to example 3, I think it is inappropriate to select covariates based on statistical significance. These covariates should be pre-specified based on clinical reasoning to avoid any possible confounding. Also, instead of discarding a large number of observations, could the authors not perform some sort of imputation for missing data?

Experimental design

Sufficient.

Validity of the findings

The comparisons are not currently sufficiently clear. In all three examples, it it would be important to compare how the estimated PDR change when using the the different models.

---

## Round 0.2 · Major Revisions

Thank you for your revisions and rebuttals, and apologies for the delay over the holiday season.

As you will see, Reviewer #1 has only a few smaller comments for you to address.

Reviewer #2 has more substantial comments. These were accidentally entered into the ‘comments for editor’ box and so I’ll reproduce them below.

Before that, I’ll make a couple of comments on my own about the simulations.

1. I appreciate the addition of the simulations. I think readers will want more information in the work itself on how the data were simulated, perhaps around Line 304.

2. I was anticipating that the parameters of the simulations would be informed by your data but still range over likely ranges of, for example, sample size, proportion of zeros, etc. so that readers have some idea of how the methods might perform using realistic data but also data that differ from the case studies.

As before, if you feel you can address my comments and those from Reviewer #2 (all of their points seem important to me), I will look forward to seeing a revised version of your manuscript along with a copy showing tracked changes and a discussion of the changes you have made in response to these comments/why you believe no changes are required for particular comments.

COMMENTS FROM REVIEWER #2 BELOW THIS POINT

Thank you for the opportunity to re-review this manuscript—Some of the issues from my original review were not adequately addressed, so I will revisit them.

1. First and foremost, when comparing the same methods, it would be ideal to compare them based on how they all estimate the same quantity (i.e., the same estimand). Can all the models considered estimate the PDR? I believe fitting a ZIP model to the data will still allow one to estimate the “implied PDR” of the fitted model, but this requires some standardization. Reply: Thank you. Yes, the same estimand has been used for estimating PDR. We have now reported the estimated PDR across models. Please review the highlighted data in Table 3.

I don’t think the authors understand the “same estimand” concept I was trying to get at. Albert et al. discusses this issue in a similar context (Stat Methods Med Res . 2014 June ; 23(3): 257–278. doi:10.1177/0962280211407800). Each model (regardless of how it is parameterized) allows one to estimate the “marginal difference in the PDR” or “marginal odds ratio of the PDR” with standardization. I have written some R code (provided at the end of this review) to show what I mean with the “Example dataset 1: CAC vs. SC procedures” and three of three of the models. Here are the results I obtain:
• With the logistic regression model, one obtains a “marginal odds ratio” estimate of 0.91 (0.59, 1.39)
• With the Poisson regression model, one obtains a “marginal odds ratio” estimate of 1.20 (0.79, 1.78)
• With the ZIP regression model, one obtains a “marginal odds ratio” estimate of 0.91 (0.69, 1.18)

• With the logistic regression model, one obtains a “marginal difference in PDR” estimate of -0.02 (-0.13, 0.08)
• With the Poisson regression model, one obtains a “marginal difference in PDR” estimate of 0.04 (-0.06, 0.14)
• With the ZIP regression model, one obtains a “marginal difference in PDR” estimate of -0.02 (-0.08, 0.04)
These estimates are actually comparable across the models! Furthermore, if clinicians wish to know how the PDRs compare with CAC vs. SC procedures, we see the advantage of using the ZIP model compared to the LR model, the estimates are much more precise (i.e., the confidence intervals are much narrower: compare (0.69, 1.18) to (0.60, 1.43) and compare (-0.08, 0.04) to (-0.13, 0.08)).

I hope the authors understand this idea and implement this for the remaining example datasets and models.

In the introduction the authors write “Moreover, clinicians are often interested in PDR estimation and comparison in the primary analysis of colonoscopic studies which cannot be obtained by simple count data models.” This is inaccurate: With standardization, we can obtain estimates of the PDR and estimates of how the PDR differs with different procedures.

“it is clear that CAC had a higher number of detected polyps compared to the SC, despite no differences in the PDR.” Such a statement can only be made if the “difference in the PDR” is estimated using each model. See comment above.

“The effect sizes obtained from the inflated models were different than standard approaches indicating the standard approaches may produce biased estimates even if the conclusions do not change.”
What effect sizes are you comparing? See comment above. While it may be an inefficient model (because of data dichotomization), the logistic regression model will not give biased estimates.

“The estimated percentage of detected polyps (56.9% vs. 48.1%) was slightly higher in the segmental protocol compared to the standard withdrawal time protocol.” I’m not sure I understand- this “estimated percentage” is based on using which model?

2. Furthermore, the authors cite Campbell but then go on to suggest that the best model is chosen based on a series of goodness of fit tests. Is this the best recommendation or should the primary model be pre-specified to avoid any model selection bias? Reply: Thank you for your insightful suggestion. We have now added this statement that models should be prespecified for primary analysis especially in interventional studies in study registration to avoid selection bias. However, a series of additional tests may be performed depending on study objectives to conduct additional sensitivity analyses for validating the robustness of findings in terms of violating assumptions. The purpose of this study is to facilitate researchers to use inflated models as the primary models for analyzing polyp data. We have added these clarifications in our revised manuscript. Please review the statistical analysis section (lines: 274-284) and the discussion section (lines: 436-438).

You currently write: “Figure 1 presents the step-by-step approach for selecting an appropriate regression model.” While this is a pragmatic approach, Campbell (2021) urge caution against this approach. Ideally, the model is selected before any data is observed. Adding a comment here explaining this issue would be beneficial.



3. Finally with respect to example 3, I think it is inappropriate to select covariates based on statistical significance. These covariates should be pre-specified based on clinical reasoning to avoid any possible confounding. Also, instead of discarding a large number of observations, could the authors not perform some sort of imputation for missing data? Reply: Yes, we adjusted variables based on their clinical significance and event-per-variable ratio to avoid instability in the estimates. We have specified all adjustments of covariates in the statistical analysis section. There is only one variable BMI that has missing data and we checked the results which did not change from the original results even after imputing this variable. Please review the statistical analysis section which provides the reasons for adjusting covariates in different analyses (lines: 285-293). Based on your recommendation, we also validated findings after imputing missing data for example dataset 3. Please review supplementary table 4.

“We only retained variables in the final models that were associated with PDR or polyp counts.” This suggests that you are still selecting covariates based on their statistical significance. This is not considered appropriate due to model selection bias issues (see concerns about stepwise selection of covariates for regression models, e.g., Steyerberg et al., 1999 J Clin Epidemiol.).
“However, we analyzed 1936 patients owing to missing data on BMI.” “example dataset 3 were also validated after imputing missing data using the multivariate normal method.” Can you provide a reference for the “multivariate normal method”? Also, please refer to the supplementary table 4 in the main text and add a comment/reference about the importance accounting for potentially non-random missingness.

“The baseline covariates included in this study were age, BMI, sex, ethnicity, fellow present, colonoscope used for adult or pediatric, preparation quality, cecal intubation, withdrawal time, procedure time, and cecal intubation time.” What qualifies as “included in this study”? Is any variable that was measured on patients at study baseline “included”? This language is unclear. I see that Avalos et al. (2020) describe these variables as “baseline characteristics of the study population according to standard and segmental withdrawal protocols”.

“However, age, sex, and total procedure time are typical predictors for PDR and therefore we adjusted for age, gender, and total procedure time while evaluating the differences between CAC and SC.” Can you provide a reference for “typical predictors”? Also, is it “sex” or “gender”?


Minor comments:

The caption and footnote for “Supplementary Table 2” must be updated to explain what is included in this table.

“Effect of implementation of fellowship program on colon polyps in the example dataset 3.” Since you “only retained variables in the final models that were associated with PDR or polyp counts”, could you clearly explain which variables ended up being included in the models. Currently I’m not sure if I understand which variables are included when you write: “The two-part analysis with the ZINB model suggested that male gender (RR=1.51, p<0.001), older age (RR=1.02, p=0.001), time of the procedure (RR=1.36, p<0.001) and moderate sedation (RR=0.62, p=0.005) were associated with the number of detected polyps but not with a presence of detected polyps. In contrast, only increases in BMI were associated with PDR (OR=1.14, p=0.008) (Table 5).”

“A p-value less than 5% was considered statistically significant” p-values are not measured in percentage points. Please correct this error.


R code for standardization:

# read and format data
peer1 <- read.csv("~/Downloads/peerj-107788-raw_datasets.csv")
peer1[,"polyp"]<-as.numeric(peer1[,"number_of_polyp_detected_1"]>0)
head(peer1)
peer1[,"study_arm_c"] <-relevel(as.factor(peer1[,"study_arm_c"]),ref="None")


# confirm numbers in the paper:
mean(peer1[peer1[,"study_arm_c"]%in%unique(peer1[,"study_arm_c"])[1],"polyp"])
# [1] 0.4123223
mean(peer1[peer1[,"study_arm_c"]%in%unique(peer1[,"study_arm_c"])[2],"polyp"])
# [1] 0.4299065
mean(peer1[,"polyp"])
# [1] 0.4211765

LR <- (glm(polyp~ study_arm_c+ total_procedure_time + age+ sex_c, family="binomial", data=peer1))
round(exp(coef(LR)[2]),2)
# 0.90
round(exp(confint(LR))[2,],2)
# 0.59 , 1.37

PR <- (glm(number_of_polyp_detected_1 ~ study_arm_c+ total_procedure_time + age+ sex_c, family="poisson", data= peer1))
round(exp(coef(PR)[2]),2)
# 1.14
round(exp(confint(PR))[2,],2)
# 0.93 , 1.41

# Supplementary Table 2
LR_2 <- (glm(polyp~ study_arm_c, family="binomial", data=peer1))
round(exp(coef(LR_2)[2]),2)
# 0.93
round(exp(confint(LR_2))[2,],2)
# 0.63 , 1.37



##### ##### ##### ##### ##### ##### #####
# function to calculate estimate of diff in PDR from LR model
LR_for_diff_PDR <- function(data, indices){
resampled_data = data[indices,]

LR <- (glm(polyp~ study_arm_c+ total_procedure_time + age+ sex_c, family="binomial", data= resampled_data))

peer1_0<- resampled_data
peer1_0[, "study_arm_c"]<-"None"
PDR_0 <- mean(rbinom(n=dim(peer1_0)[1], size=1, prob=predict(LR, type="response", newdata= peer1_0))>0)

peer1_1<- resampled_data
peer1_1[, "study_arm_c"]<-"CAP"
PDR_1 <- mean(rbinom(n=dim(peer1_1)[1], size=1, prob=predict(LR, type="response", newdata= peer1_1))>0)

odds_PDR_0 <- PDR_0/(1-PDR_0)
odds_PDR_1 <- PDR_1/(1-PDR_1)

diff_PDR <- PDR_1 - PDR_0
return(diff_PDR)}

##### ##### ##### ##### ##### ##### #####
# function to calculate estimate of diff in PDR from PR model
PR_for_diff_PDR <- function(data, indices){
resampled_data = data[indices,]

PR <- (glm(number_of_polyp_detected_1 ~ study_arm_c+ total_procedure_time + age+ sex_c, family="poisson", data=resampled_data))

peer1_0<- resampled_data
peer1_0[, "study_arm_c"]<-"None"
PDR_0 <- mean(rpois(n=dim(peer1_0)[1],lambda=predict(PR, type="response", newdata= peer1_0))>0)

peer1_1<- resampled_data
peer1_1[, "study_arm_c"]<-"CAP"
PDR_1 <- mean(rpois(n=dim(peer1_1)[1],lambda=predict(PR, type="response", newdata= peer1_1))>0)

diff_PDR <- PDR_1 - PDR_0
return(diff_PDR)}

##### ##### ##### ##### ##### ##### #####
# function to calculate estimate of diff in PDR from ZIP model
library(VGAM)
library(pscl)
# https://stats.stackexchange.com/questions/189005/simulate-from-a-zero-inflated-poisson-distribution
ZIP_for_diff_PDR <- function(data, indices){
resampled_data = data[indices,]

ZIP <- zeroinfl(polyp ~ study_arm_c+ total_procedure_time + age+ sex_c| study_arm_c+ total_procedure_time + age+ sex_c,
dist = 'poisson',
data = peer1)

peer1_0<- resampled_data
peer1_0[, "study_arm_c"]<-"None"
n <- dim(peer1_0)[1]
p <- predict(ZIP, newdata= peer1_0, type = "zero")
lambda <- predict(ZIP, newdata= peer1_0, type = "count")
PDR_0 <- mean(rzipois(n, lambda = lambda, pstr0 = p)>0)

peer1_1<- resampled_data
peer1_1[, "study_arm_c"]<-"CAP"
n <- dim(peer1_1)[1]
p <- predict(ZIP, newdata= peer1_1, type = "zero")
lambda <- predict(ZIP, newdata= peer1_1, type = "count")
PDR_1 <- mean(rzipois(n, lambda = lambda, pstr0 = p)>0)

diff_PDR <- PDR_1 - PDR_0
return(diff_PDR)}


##### ##### ##### ##### ##### ##### #####
# Results for diff in PDR:
library(boot)
set.seed(123)

LR_for_diff_PDR_boot1 <- boot(data= peer1, statistic= LR_for_diff_PDR, R= 10000)
round(c(quantile(LR_for_diff_PDR_boot1$t, c(0.5,0.025,0.975))),2)
# -0.02 -0.13 0.08

PR_for_diff_PDR_boot1 <- boot(data= peer1, statistic= PR_for_diff_PDR, R= 10000)
round(c(quantile(PR_for_diff_PDR_boot1$t, c(0.5,0.025,0.975))),2)
# 0.04 -0.06 0.14

ZIP_for_diff_PDR_boot1 <- boot(data= peer1, statistic= ZIP_for_diff_PDR, R= 10000)
round(c(quantile(ZIP_for_diff_PDR_boot1$t, c(0.5,0.025,0.975))),2)
# -0.02 -0.08 0.04


##### ##### ##### ##### ##### ##### #####
# function to calculate estimate for OR of PDR from LR model
LR_for_OR_PDR <- function(data, indices){
resampled_data = data[indices,]

LR <- (glm(polyp~ study_arm_c+ total_procedure_time + age+ sex_c, family="binomial", data= resampled_data))

peer1_0<- resampled_data
peer1_0[, "study_arm_c"]<-"None"
PDR_0 <- mean(rbinom(n=dim(peer1_0)[1], size=1, prob=predict(LR, type="response", newdata= peer1_0))>0)

peer1_1<- resampled_data
peer1_1[, "study_arm_c"]<-"CAP"
PDR_1 <- mean(rbinom(n=dim(peer1_1)[1], size=1, prob=predict(LR, type="response", newdata= peer1_1))>0)

odds_PDR_0 <- PDR_0/(1-PDR_0)
odds_PDR_1 <- PDR_1/(1-PDR_1)

OR_PDR <- odds_PDR_1/odds_PDR_0
return(OR_PDR)}

##### ##### ##### ##### ##### ##### #####
# function to calculate estimate for OR of PDR from PR model
PR_for_OR_PDR <- function(data, indices){
resampled_data = data[indices,]

PR <- (glm(number_of_polyp_detected_1 ~ study_arm_c+ total_procedure_time + age+ sex_c, family="poisson", data=resampled_data))

peer1_0<- resampled_data
peer1_0[, "study_arm_c"]<-"None"
PDR_0 <- mean(rpois(n=dim(peer1_0)[1],lambda=predict(PR, type="response", newdata= peer1_0))>0)

peer1_1<- resampled_data
peer1_1[, "study_arm_c"]<-"CAP"
PDR_1 <- mean(rpois(n=dim(peer1_1)[1],lambda=predict(PR, type="response", newdata= peer1_1))>0)

odds_PDR_0 <- PDR_0/(1-PDR_0)
odds_PDR_1 <- PDR_1/(1-PDR_1)

OR_PDR <- odds_PDR_1/odds_PDR_0
return(OR_PDR)}

##### ##### ##### ##### ##### ##### #####
# function to calculate estimate for OR of PDR from ZIP model
library(VGAM)
library(pscl)
# https://stats.stackexchange.com/questions/189005/simulate-from-a-zero-inflated-poisson-distribution
ZIP_for_OR_PDR <- function(data, indices){
resampled_data = data[indices,]

ZIP <- zeroinfl(polyp ~ study_arm_c+ total_procedure_time + age+ sex_c| study_arm_c+ total_procedure_time + age+ sex_c,
dist = 'poisson',
data = peer1)

peer1_0<- resampled_data
peer1_0[, "study_arm_c"]<-"None"
n <- dim(peer1_0)[1]
p <- predict(ZIP, newdata= peer1_0, type = "zero")
lambda <- predict(ZIP, newdata= peer1_0, type = "count")
PDR_0 <- mean(rzipois(n, lambda = lambda, pstr0 = p)>0)

peer1_1<- resampled_data
peer1_1[, "study_arm_c"]<-"CAP"
n <- dim(peer1_1)[1]
p <- predict(ZIP, newdata= peer1_1, type = "zero")
lambda <- predict(ZIP, newdata= peer1_1, type = "count")
PDR_1 <- mean(rzipois(n, lambda = lambda, pstr0 = p)>0)

odds_PDR_0 <- PDR_0/(1-PDR_0)
odds_PDR_1 <- PDR_1/(1-PDR_1)

OR_PDR <- odds_PDR_1/odds_PDR_0
return(OR_PDR)}


##### ##### ##### ##### ##### ##### #####
# Results for OR:
set.seed(123)

LR_for_OR_PDR_boot1 <- boot(data= peer1, statistic= LR_for_OR_PDR, R= 10000)
round(c(quantile(LR_for_OR_PDR_boot1$t, c(0.5,0.025,0.975))),2)
# 0.91 0.59 1.39

PR_for_OR_PDR_boot1 <- boot(data= peer1, statistic= PR_for_OR_PDR, R= 10000)
round(c(quantile(PR_for_OR_PDR_boot1$t, c(0.5,0.025,0.975))),2)
# 1.20 0.79 1.78

ZIP_for_OR_PDR_boot1 <- boot(data= peer1, statistic= ZIP_for_OR_PDR, R= 10000)
round(c(quantile(ZIP_for_OR_PDR_boot1$t, c(0.5,0.025,0.975))),2)
# 0.91 0.69 1.18

Reviewer 1 ·

Basic reporting

no comment

Experimental design

no comment

Validity of the findings

no comment

Additional comments

In the abstract, authors have concluded that zero-inflated models perform better in both empirical and simulation analyses, but only provided the empirical results in the results. Would be great adding something for simulations.

A two-sided p-value, isn't it?

Please write "Stata v17.0 (StataCorp, Lakeway Drive, College Station, Texas, USA)....".

Please maintain the uniformity when reporting the RR/OR in the results section. These are reported with 95%CI at some places, and just with the p-values at some places. Kindly report with 95% CI throughout the manuscript.

Reviewer 2 ·

Basic reporting

Sufficient.

Experimental design

Sufficient.

Validity of the findings

Sufficient.

---

## Round 0.3 · Minor Revisions

Thank you very much for your revisions and responses. As you can see, Reviewer #2 has a small suggestion for you to consider about Supp Table 2 and connected discussion. I’ll leave this point for you to decide on, but I think that their suggestion has considerable merit.

I’ll also note some likely typos and a few other comments for you to address before I can accept the manuscript. Providing you consider Reviewer #2’s comment (you are not required to make changes in light of it, but I think it would improve your work) and address my points below (or explain why the current text is intended), and nothing new comes to light, I hope to be delighted to accept your manuscript after resubmission. I hope that you can bear with one, I hope, final round of revisions.

As a general comment, in places the sentence structure or phrasing could be improved, although the intent mostly seems clear (with some exceptions that I’ve noted below). I suggest that very careful proof-reading could address this.

Line 25: When talking about these techniques in general, I think “potentially inaccurate” is more reasonable. In some cases, the question may be related to dichotomized data, for example, and count data models could be ‘inaccurate’ for that question. See also Line 44 where again I’d suggest “potentially inaccurate” as it’s likely that for some inclusion/exclusion criteria applied to some populations there will not be zero-inflation. I suggest that the word ‘inaccurate’ be carefully considered throughout your work so that you avoid over-stating this point. See Line 69 and throughout the body of the manuscript for this point.

Line 32: Missing space in “Poisson regression(PR),” (c.f. other similar uses on the same line). Worth checking throughout for similar.

Line 76: While I’m not a fan of no space before the opening parenthesis of a reference, you’ve used that approach until here. You go back to no spaces until Line 83 when the space reappears. The formatting of references should be consistent throughout.

Lines 84–87: This sentence seems awkwardly constructed. Please check that the first ‘or’ on Line 86 is intended. Check also that the plural ‘models’ on Line 87 is not intended to be linked to the singular ‘a’ on Line 85.

Line 90: Do you need a comma before “which” here? I won’t comment on commas throughout but a read through checking that they are included when needed and not otherwise should be useful.

Lines 101–102: A reference for this sentence (“As per value-based biostatistics practice…”) would be very welcome.

Lines 107–109: There’s always a risk of such absolute statements, rather than hedging with ‘to the best of our knowledge’ and similar, and I think that https://doi.org/10.1002/sim.10210 (submitted 28 July 2023 and first published online 15 September 2024, about a month prior to your original submission on 24 Oct 2024, so very easy to not have picked up then) would qualify as a counter-example here. Consider also Lines 535–537 and perhaps elsewhere.

Line 114: Perhaps “over-, under-, or equi-dispersion outcome data” for clarity? See also Lines 276–277 and elsewhere.

Line 161: I’m not sure that the ‘the’ in “that is the no polyps” is intended/needed. Check for any missing or spurious articles (‘the’, ‘a’, and ‘an’)

Lines 167–168: I’m not sure that readers will appreciate why you’ve defined i as being from 1 to n when n remains undefined. This is a little pedantic, but i should be defined when used. See also around Lines 182, 209, 215, 242, 247, and perhaps elsewhere.

Line 176: It would be helpful to some readers to note that this is the incidence rate ratio (IRR). I suspect that some readers will recognise this term in particular. See also Line 215 and elsewhere.

Line 180: I think you mean “the expected count of polyps” not the “of the absence”. Also note that this requires exponentiating rather than being interpretable directly.

Line 210: “The inverse of OR provides the effect of the corresponding covariate on the detected polyps.” could be seen as describing a reciprocal. Do you mean the back-transformed OR here?

Line 213: Do you mean given “no excess zero” here?

Line 245: Do you mean given “polyps” here?

Lines 270–272: I don’t think all readers will follow your logic here.

Line 298: Do you mean “adjusted” here: “Some variables were adjusted or removed”? If so, readers might want to know what ‘adjusted’ means in this context.

Line 306: Do you mean that this was determined “then” or “also”?

Line 393: Note that MI only accounts for MCAR and MAR conditional on variables in the imputation model, i.e., doesn’t in itself accommodate non-random missingness. It can be a basis for accommodating these, but it’s not sufficient alone.

Line 399: Readers might want to know the comparison reported by the “older age” results (this could be per year, per number of years, or compared to a reference group and the text would not help a reader understand which applied). This is clear in Table 5, but the text should be readable without needing to check the tables.

Line 399: Similarly, some readers will want to know the comparison reported by the “time of the procedure” results. Same for BMI on Line 402.

Reviewer 2 ·

Basic reporting

No further comments.

Experimental design

No further comments.

Validity of the findings

Supplemental Table 2 provides a lot of value and I would suggest that the authors move it to the main text. It is interesting to compare how the width of the confidence interval differs across the different models allowing us to ask ourselves "which model provides the most precise estimate?" This is something that I think should be discussed in the paper, but is only a suggestion.

Additional comments

No further comments.

---

## Round 0.4 · accepted · Accept

Thank you for your revisions. I am delighted to accept your manuscript!